# Ontogenetic Structure and Temporal Patterns of Summer Ichthyoplankton in Upper Course of the Xijiang River, SW China

Minghui Gao [1], Zhiqiang Wu [1,2,3,*], Xichang Tan [4], Liangliang Huang [2], Jie Feng [2] and Saeed Rad [2]

1 College of Life Science and Technology, Guangxi University, Nanning 350000, China; gmhxxzj@126.com
2 College of Environmental Science and Engineering, Guilin University of Technology, Guilin 541000, China; llhuang@glut.edu.cn (L.H.); Jeffreywh@163.com (J.F.); saeedrad1979@gmail.com (S.R.)
3 School of Marine Sciences, Guangxi University, Nanning 350000, China
4 Bureau of Hydrology and Water Resources, Pearl River Conservancy Commission of Ministry of Water Resources, Guangzhou 510000, China; jimtxc@hotmail.com
* Correspondence: wuzhiqiang@glut.edu.cn; Tel.: +86-187-7718-0208

**Abstract:** The summer ichthyoplankton characteristics in the Laibin section of the Xijiang River were analyzed based on a survey during summer 2017. The ontogenetic structure and temporal patterns of ichthyoplankton and the correlation between environmental parameters and the temporal patterns were investigated. A total of 10,665 eggs and 447 larval belonging to four orders, ten families, and 28 species were collected. According to the flood regime, summer is divided into three periods (pre-flood, flood period, and post-flood). Ichthyoplankton proved to be heterogeneous between periods in summer with differences in the composition and abundance. The assemblages were distinguished by multiple analytical tools, and presented a chronological pattern of marked variability in composition of the species between the periods, and under the strong influence of flood. The assemblages were mainly represented by eggs of *S. wui Fang* and *S. robusta* in the pre-flood period, *S. argentatus* and *S. macrops* in the flood period, and *H. leucisculus* and *S. curriculus* in the post-flood period, while, the larval occurred mainly in the flood period. Understanding these temporal patterns of the upper course of the Xijiang River is useful for the recruitment of fish resources and conservation of fish community diversity.

**Keywords:** ichthyoplankton; temporal patterns; spawning; Xijiang River

## 1. Introduction

Due to the influence of human factors, the global biodiversity is declining by an alarming rate, the species extinction gradient today is at least tens to hundreds of times faster than the average rate of the past 10 million years [1]. For global freshwater fish diversity, dam construction, overfishing, biological invasions, and climate change are the main reasons for the decline [2]. Among these causes, dam construction is considered to be the main cause for the degradation of the aquatic ecosystem and the decline in freshwater fish diversity. It is due to the fact that dam construction leads to the loss of fish habitat, the destruction of spawning grounds, cutting off of migratory routes [3], and affecting genetic differentiation of freshwater organisms [4], etc. Globally, 77% of rivers over 1000 km in length, are fragmented due to dams, reservoirs, and other facilities [5]. Therefore, it is very necessary to develop practical and effective diversity conservation strategies for fragmented rivers. Understanding the population and dynamics of fish is basic for developing the strategies [6]. Since ichthyoplankton abundance and species composition largely reflect the population and dynamics of the adult fish. Therefore, ichthyoplankton resources, surveys, and recruitment dynamics analysis are effective means to obtain information on populations [7].

In recent years, there has been increasing interest in the annual and seasonal variations in the occurrence patterns of ichthyoplankton. The fact that research focuses on why

patterns occur in certain periods has important practical significance [8], both for the protection and utilization of fishery resources [9]. In addition, the temporary models have been at the center of much research on marine and freshwater ecosystems around the world [8–11].

It is well known that temporal patterns of ichthyoplankton occurrence vary between different environments, and depends on the local environmental factors [12]. Studies have demonstrated that photoperiod, water temperature, and food availability are the main factors affecting recruitment dynamics in temperate regions [13]. In tropical regions, increases in water level, precipitation, and electrical conductivity are considered determinants of their reproductive patterns [14,15]. In the subtropical region, however, the occurrence patterns are closely related to variations in flood and water temperature, and for most fish species, spawning occurs in periods of flood [7]. The previous studies have mainly focused on annual and seasonal variations in occurrence patterns of ichthyoplankton, but few studies on the occurrence patterns of ichthyoplankton during the main spawning season. To develop the conservation strategies for fragmented rivers, such as ecological regulation, more accurate and detailed reproductive information is needed. The Xijiang basin is located at subtropical latitudes. The warm climate, abundant rainfall, frequent floods, and complex river system provide plenty of habitats and a rich feed for fish [16,17]. Two hundred and sixteen fish species have been recorded in Xijiang, 30 of which are endemic [18]. Additionally, the spawning grounds of most fish species of economic importance are distributed in its middle and upper regions [16,19]. In the upper and middle reaches of Xijiang, eleven dams have been designated, with the first completed in 1980 and the final dam (Datengxia) intended to be operational by 2023. Hence, fish diversity and resources are under threat, and the composition of fish species has been significantly affected by the construction of cascade reservoirs [20].

In tropical rivers, the temporal pattern of ichthyoplankton occurrence is seasonal and for most fish species spawning occurs during flood events [13]. Studies in the lower and middle reaches of Xijiang show that the highest larval densities are observed between May and July. Moreover, the temporal patterns of larval occurrences are very related to flood and temperature [7,19]. In the upper reaches of Xijiang, however, studies on fish reproduction are scarce, and little information is available related to the temporal patterns. Moreover, the abiotic variables which influence the reproduction of the existing fish communities are unknown at these latitudes under the dam control.

At present, the Laibin section is the last dam-free section of the upper Xijiang River. This study aims to (1) describe the temporal patterns of ichthyoplankton occurrence during the summer in this section, (2) determining the underlying influences affecting ichthyoplankton by analyzing the relationships between the temporal patterns of ichthyoplankton occurrence and environmental factors under the dam control.

## 2. Materials and Methods

### 2.1. Study Area

Xijiang River is the largest tributary of Pearl River, at 2214 km in length and a catchment area of 353,120 km$^2$ and means annual discharge of $2.24 \times 10^{11}$ m$^3$. The upper Xijiang River is located in a geotectonically complex karst area with mountains and valleys on either side, curved channels, beaches, underground streams, and karst caves [16]. It has a subtropical climate, with a mild climate and abundant rainfall. Seasonally, high flows occur in June and July during summer. Since the 1980s, the landscape and the hydrodynamics of the basin have been modified by the construction of various hydroelectric power plants (HEP). This study was carried out in the Laibin section under the influence of the Qiaogong HEP, covering an area of approximately 80 kms of the upper Xijiang River (Figure 1).

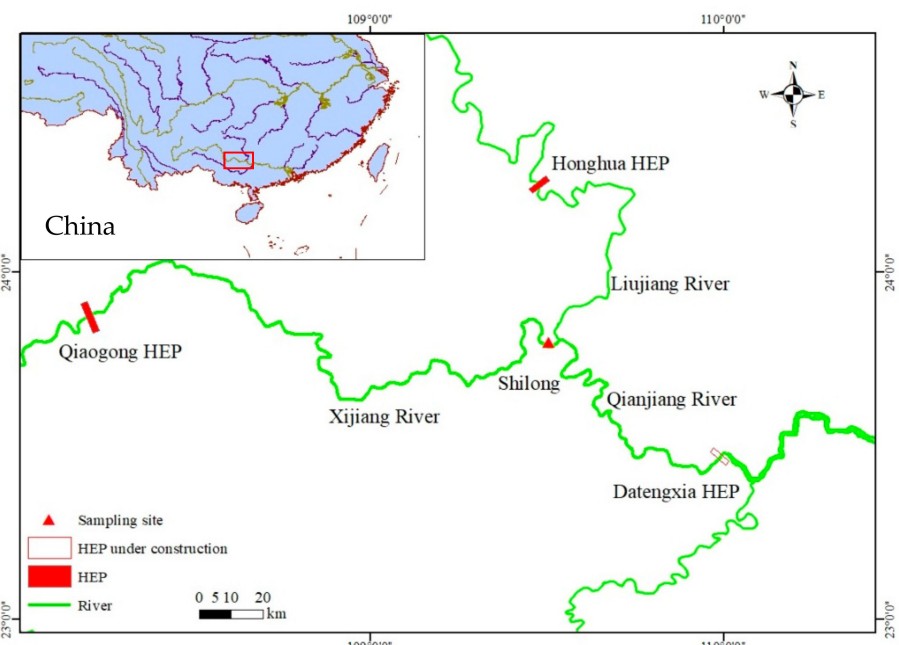

**Figure 1.** Study area.

*2.2. Sampling and Data Collection*

Samples were collected daily during the spawning season (from April to August) in 2017. The Shilong (109°32′6″ E, 23°47′36″ N) sampling site was set up to collect ichthyoplankton. Sampling was conducted at the sampling station once each day. The duration of the sample collection was 1 h (06:00–07:00) against the current. For this purpose, ichthyoplankton samples were collected with drift nets (total length 5 m; rectangular iron opening/mouth 1.0 × 1.5 m, and a mesh net size of 0.5 mm attached to a 0.8 × 0.4 × 0.4 m filter collection bucket) [19]. The nets were deployed at the left bank on the sampling site, a mechanical flow meter was attached to the mouth of the net to calculate the volume of water filtered into the nets.

The collected ichthyoplankton samples were counted before being sorted to the lowest possible taxonomic level at each station according to morphological characteristics [21,22]. Eggs that could not be identified were placed in an aerated tank (30 cm diameter and 40 cm height) at 20–25 °C for 1 week or more until the species could be identified, with each tank stocked with 40–50 eggs. Real-time water temperature and Dissolved Oxygen (DO) data were collected using a dual input multi parameter digital analyzer (HACH HQ40d). Data on water transparency were measured using a Secchi disk (Wuhan HENGLING Technology Co., Wuhan, China.). Data on discharge and water levels were from the Pearl River Water Conservancy Commission's website (http://www.pearlwater.gov.cn (accessed on 16 April to 30 August 2017)).

*2.3. Data Analysis*

Drift densities were standardized for a volume of 100 m$^3$ of filtered water (no. of individuals 100 m$^{-3}$). The water volume was calculated based on sampling duration (s), mean current velocity at the net entrance (m/s), and area of the net entrance (m$^2$). In order to verify the existence of an environmental gradient, the Principal Component Analysis (PCA) was used. Axes with eigenvalues greater than 1.0 were used for interpretation according to the Kaiser-Guttman criterion [23]. Only limnological variables with score coefficients > 0.4 were considered biologically important [24]. In this study, according to flood patterns, summer is divided into three periods (pre-flood period, flood period, and post-flood period). Abundance patterns were analyzed for each period based on the daily dataset.

The temporal variation in ichthyoplankton assemblage composition was verified by a Non-Metric Multidimensional Scaling (NMDS) analysis performed with the density [25]. An Analysis of Similarity (ANOSIM) was used to evaluate possible significant differences in the composition of ichthyoplankton. To emphasize the main taxa responsible for the formation of the delineated groups, a Similarity Percentage (SIMPER) was applied [26].

Additional attention was paid to the relationship between the occurrence pattern of ichthyoplankton and environmental factors and to find out which environmental variables were most related to species occurrence based on abundance. First, the Detrended Correspondence Analysis (DCA) was used to analyze the density of species and hydrological factor data, then select the appropriate method according to the Length of each Axis's Gradients (LGA). Due to the sensitivity of DCA to rare species, dominant species were selected for further analysis [27]. For LGA < 3, Redundancy Analysis (RDA) was used; when LGA > 4, Canonical Correlation Analysis (CCA) was used; and for 3 < LGA < 4, both analysis methods are applicable [28]. In this study, the RDA was employed to model the relationship between the occurrence pattern and environmental factors, as gradient length < 3. The SIMPER analysis was performed using the PRIMER 5 software. PCA, NMDS analysis, ANOSIM, DCA, and RDA were performed via the R Statistical Software.

## 3. Results

### 3.1. Environmental Factors

From 16 April to 30 August 2017, the water discharge rate varied from 1310 to 8520 m$^3$ s$^{-1}$, and the water level ranged from 63.08 to 76.46 m at the Qianjiang Hydrologic Station, 30 km upstream from our sampling site. Floods mainly occurred from 16 June to 27 July in the summer. According to the occurrence mode of the flood, summer was divided into three periods: Pre-flood period (16 April to 15 June), flood period (16 June to 27 July), and post-flood period (28 July to 30 August) (Figure 2).

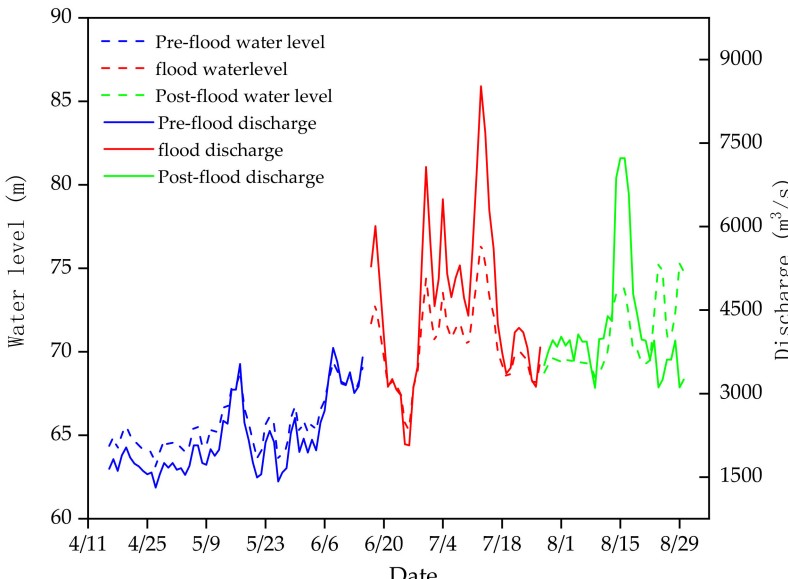

**Figure 2.** Variation in discharge and water level in the Laibin section of the Pearl River from 16 April to 30 August 2017.

Temporal variations of five environmental factors are shown in Table 1 and Figure 3. The water discharge rate, water level, and temperature were highly correlated with each other and negatively associated with transparency. These are linked to PC1 which accounts for 66.3% of the total variance. DO is linked to axis 2, explaining 15.82% of the total variance. All environmental factors showed timely gradients along PC1, corresponding to the flood characteristics by the high river discharge rate, high water level, low transparency, and low

DO to the pre-flood and post-flood period. The water temperature was gradually rising in summer and its influence was mainly reflected on the post-flood period.

**Table 1.** Temporal variations in range and mean ± standard deviation (SD) values for water temperature, discharge, water levels, dissolved oxygen (DO), and water transparency in the upper Xijiang in summer 2017.

| Period | Water Temperature (°C) | | Discharge (m³/s) | | Water Levels (m) | | DO (mg/L) | | Water Transparency (cm) | |
|---|---|---|---|---|---|---|---|---|---|---|
| | Range | Mean ± SD | Range | Mean ± SD | Range | Mean ± SD | Range | Mean ± SD | Range | Mean ± SD |
| **Pre-flood** | 15–24.5 | 19.17 ± 4.07 | 1310–3070 | 1836.67 ± 346.67 | 63.08–67.88 | 64.82 ± 0.96 | 6.32–6.83 | 6.55 ± 0.16 | 100–125 | 111.97 ± 7.60 |
| **Flood** | 22.6–25.8 | 23.63 ± 0.71 | 1420–8520 | 3707.78 ± 1579.71 | 63.5–76.46 | 68.77 ± 2.93 | 6.12–7.02 | 6.56 ± 0.26 | 20–125 | 70.44 ± 27.32 |
| **Post-flood** | 24.3–27.2 | 26.15 ± 0.91 | 3100–7230 | 4240 ± 1095.30 | 68.46–75.29 | 70.48 ± 1.89 | 5.64–5.98 | 5.78 ± 0.12 | 54–105 | 76.25 ± 13.99 |

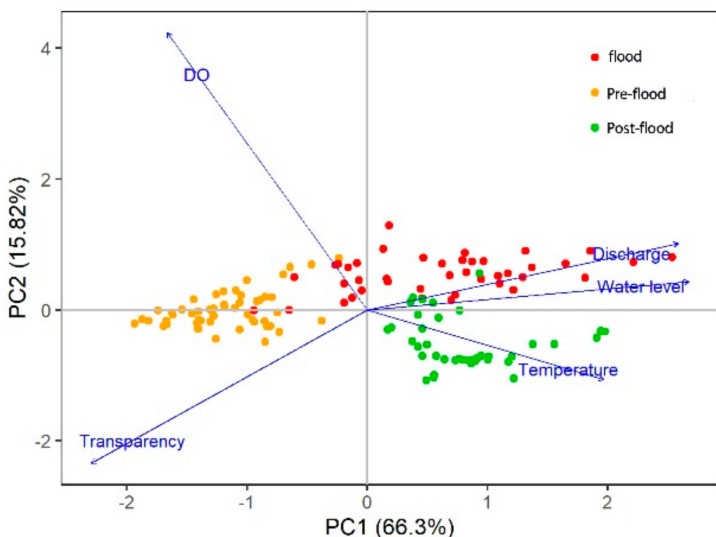

**Figure 3.** Principal component analysis (PCA) analysis among limnological variables in the Laibin section of the Pearl River during the summer of 2017.

### 3.2. Species Composition of Ichthyoplankton

A total of 134 samples were collected over the summer. In these samples, 10,665 eggs and 447 larval were identified. All the samples were found to belong to four orders, ten families, and 28 species, including parts of eggs that could not be identified (1.41%). Ichthyoplankton, which belongs to the Cyprinidae family, represented the largest number of species (15) and comprised 53.6% of the total number of species with more than 66.46% of the captured and identified individuals. It is registered as the largest species richness and composes the groups of the highest numerical abundance. According to the types of eggs, there are nine species of pelagic eggs, 12 species of adhesive eggs, and seven species of other types of eggs.

The Index of Relative Importance (IRI) was used to discuss the dominant species. Species with an index greater than 1000 and a range from 100 to 1000 were considered a dominant species and common species, respectively. During the pre-flood period, 23 species had been identified. The dominant species were *S. robusta*, *S. argentatus*, and *S. wui*, while for the common species *S. curriculus*, *H. leucisculus*, *C. molitorella*, *S. macrops*, and *S. dabryi* were mainly included. During the flood period, 24 species had been identified, in which the dominant species was only *Squalidus argentatus*, and the common species mainly included *S. pulchra*, *S. curriculus*, *H. leucisculus*, *C. erythropterus*, *S. macrops*, *S. dabryi*, *C. molitorella*, *O. salsburyi*, and *S. wui*. During the pre-flood period, however, 25 species had been identified, in which the dominant species were *H. leucisculus*, *S. argentatus*, and *S. wui*, but the common species mainly included *S. robusta*, *S. pulchra*, *S. zebra*, *S. curriculus*, *S. macrops*, and *C. molitorella* (Table 2).

**Table 2.** Species composition and mean density (no. of larvae per 100 m³) of the ichthyoplankton collected in the upper Xijiang in the summer of 2017.

| Taxon | Egg Type | Life Stages of Fish | Pre-Flood | | Flood | | Post-Flood | |
|---|---|---|---|---|---|---|---|---|
| | | | Density (ind.·100⁻³) | IRI | Density (ind.·100⁻³) | IRI | Density (ind.·100⁻³) | IRI |
| **SALMONIFORMES** | | | | | | | | |
| **Salangidae** | | | | | | | | |
| *Neosalanx taihuensis* Chen, 1956 | O | larval | 0.0012 | 0.0144 | 0.0106 | 0.0122 | 0.0059 | 2.7429 |
| CYPRINIFORMES | | | | | | | | |
| **Cobitidae** | | | | | | | | |
| *Sinibotia robusta* (Wu, 1939) | P | eggs | 1.4776 | 1048.72 ** | 0.3131 | 45.77 | 0.1475 | 101.98 * |
| | | larval | 0.0044 | | 0.0078 | | 0.0216 | |
| *Sinibotia pulchra* (Wu, 1939) | P | eggs | 0.3240 | 98.84 | 0.3784 | 165.56 * | 0.4778 | 753.54 * |
| *Sinibotia zebra* (Wu, 1939) | P | eggs | 0.1680 | 53.65 | 0.1668 | 66.60 | 0.1193 | 135.01 * |
| | | larval | 0.0011 | | | | 0.0072 | |
| *Squaliobarbus curriculus* (Richardson, 1846) | P | eggs | 0.3596 | 167.79 * | 0.8156 | 508.90 * | 0.4444 | 845.43 * |
| | | larval | 0.0251 | | 0.0106 | | 0.1079 | |
| *Hemiculter leucisculus* (Basilewsky, 1855) | A | eggs | 1.1124 | 794.57 * | 0.9909 | 774.69 * | 0.7626 | 1261.47 ** |
| | | larval | 0.0284 | | 0.0026 | | 0.0144 | |
| *Chanodichthys erythropterus* (Basilewsky, 1855) | A | eggs | 0.4861 | 225.37 | 0.3904 | 224.85 * | 0.0283 | 15.79 |
| | | larval | | | 0.0321 | | | |
| *Sinibrama macrops* (Günther, 1868) | P | eggs | 0.7246 | 274.67 * | 1.1307 | 731.14 * | 0.1159 | 118.28 * |
| | | larval | 0.0016 | | | | | |
| *Hypophthalmichthys nobilis* (Richardson, 1845) | A | larval | | | 0.0023 | 0.09 | | |
| *Squalidus argentatus* (Sauvage and Dabry de Thiersant, 1874) | P | eggs | 3.8542 | 2922.07 ** | 5.9786 | 4440.72 ** | 2.6720 | 1481.5 ** |
| | | larval | 0.0087 | | 0.0086 | | 0.0504 | |
| *Saurogobio dabryi* Bleeker, 1871 | P | eggs | 0.2826 | 124.16 * | 0.3370 | 179.50 * | 0.0201 | 13.98 |
| *Rhodeus ocellatus* (Kner, 1866) | O | larval | 0.0035 | 0.08 | 0.0272 | 4.65 | 0.0059 | 2.47 |
| *Acrossocheilus kreyenbergii* (Regan, 1908) | A | larval | 0.0059 | 0.07 | 0.0686 | 19.56 | 0.0035 | 1.48 |
| *Onychostoma gerlachi* (Peters, 1881) | A | larval | | | | | 0.0177 | 7.41 |

**Table 2.** *Cont.*

| Taxon | Egg Type | Life Stages of Fish | Pre-Flood Density (ind.·100⁻³) | Pre-Flood IRI | Flood Density (ind.·100⁻³) | Flood IRI | Post-Flood Density (ind.·100⁻³) | Post-Flood IRI |
|---|---|---|---|---|---|---|---|---|
| *Cirrhinus molitorella* (Valenciennes, 1844) | A | eggs | 0.1326 | 22.51 | 0.4508 | 281.44 * | 0.2873 | 382.47 * |
| | | larval | 0.0093 | | 0.0026 | | 0.0072 | |
| *Osteochilus salsburyi* Nichols and Pope, 1927 | O | eggs | 0.1750 | 32.03 | 0.5854 | 311.76 * | 0.0981 | 81.95 |
| *Ptychidio jordani* Myers, 1930 | O | larval | | | | | 0.0141 | 4.61 |
| *Cyprinus carpio* Linnaeus, 1758 | A | eggs | 0.0213 | 2.59 | | | | |
| *Carassius auratus* (Linnaeus, 1758) | A | eggs | 0.0130 | 0.95 | | | | |
| **Homalopteridae** | | | | | | | | |
| *Sinogastromyzon wui* Fang, 1930 | P | eggs | 4.0345 | 2854.91 ** | 1.6949 | 741.37 * | 1.0668 | 1632.92 ** |
| | | larval | | | | | 0.0072 | |
| **SILURIFORMES** | | | | | | | | |
| **Clariidae** | | | | | | | | |
| *Clarias fuscus* (Lacepède, 1803) | A | larval | | | 0.0094 | 1.08 | 0.0012 | 0.5485 |
| **Bagridae** | | | | | | | | |
| *Tachysurus fulvidraco* (Richardson, 1846) | A | larval | 0.0023 | 0.06 | 0.0485 | 11.98 | 0.0047 | 2.19 |
| *Hemibagrus guttatus* (Lacepède, 1803) | A | larval | 0.0012 | 0.01 | 0.0130 | 1.24 | 0.0059 | 2.74 |
| **Perciformes** | | | | | | | | |
| **Serranidae** | | | | | | | | |
| *Siniperca chuatsi* (Basilewsky, 1855) | P | larval | 0.0047 | 0.11 | 0.0283 | 6.47 | 0.0059 | 2.47 |
| **Cichlidae** | | | | | | | | |
| *Oreochromis mossambicus* (Peters, 1852) | O | larval | 0.0047 | 0.11 | 0.1029 | 33.26 | 0.0070 | 3.62 |
| *Oreochromis niloticus* (Linnaeus, 1758) | O | larval | 0.0047 | 0.17 | 0.0709 | 22.94 | 0.0047 | 2.19 |
| **Gobiidae** | | | | | | | | |
| *Rhinogobius giurinus* (Rutter, 1897) | A | eggs | | 0.01 | | 1.57 | 0.0812 | 3.95 |
| | | larval | 0.0011 | | 0.0018 | | 0.0070 | |
| **Mastacembelidae** | | | | | | | | |
| *Mastacembelus armatus* (Lacepède, 1800) | O | larval | | | 0.0532 | 14.17 | 0.0047 | 2.19 |

### 3.3. Temporal Variation of Ichthyoplankton

The species composition of the eggs and larvae is shown in Figure 4. The composition differed during the different periods. The proportions of species in the Leuciscinae, the Cultrinae, and the Gobiobotinae increased to the occurrence of flood, while those of species in the Botiinae and the Homalopteridae decreased (Figure 4).

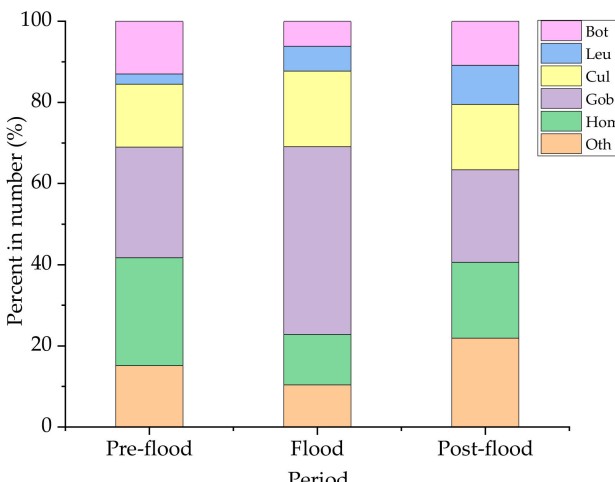

**Figure 4.** Percentages of eggs and larvae in dominant groups among samples collected from the Laibin section of Xijiang in the summer of 2017. Bot: Botiinae; Leu: Leuciscinae; Cul: Cultrinae; Gob: Gobiobotinae; Hom: Homalopteridae; Oth: Others.

The composition of fish larvae assemblages was significantly different (ANOSIM; global R = 0.540; $p < 0.001$) among the sampled periods, revealing the influence of the environmental gradient on the ichthyoplankton community structuring (Figure 5). SIMPER analysis indicated high mean dissimilarity values among the three sampled periods (>80%) and which taxa most influenced the differences between periods (Table 3).

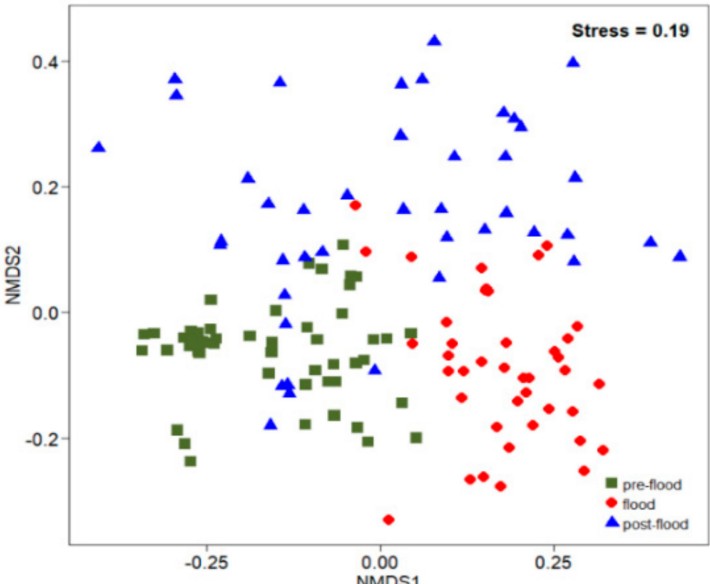

**Figure 5.** Ordering resulting from the analysis of non-metric multidimensional scaling (NMDS) on the density of ichthyoplankton based on Euclidean distance at the Laibin section of Xijiang in the summer of 2017.

**Table 3.** Paired comparisons of ichthyoplankton assemblage between periods: Pre-flood, flood, post-flood. The values correspond to the percentage of dissimilarity (similarity percentage (SIMPER) analysis) of the taxa that together contributed to more than 80% of the mean dissimilarity between the periods. The species that contribute the most to the differences are in bold. The grayscale corresponds to changes in the relative abundance of larvae in each section.

| Taxons | Pre-Flood Versus Flood | Flood Versus Post-Flood | Pre-Flood Versus Post-Flood |
|---|---|---|---|
| *Sinogastromyzon wui* | **20.41** | **31.80** | **15.36** |
| *Squalidus argentatus* | **30.06** | **16.13** | **32.61** |
| *Sinibotia robusta* | **6.49** | **9.87** | 4.92 |
| *Sinibrama macrops* | 9.15 | | **9.79** |
| *Hemiculter leucisculus* | | **9.64** | 5.76 |
| *Squaliobarbus curriculus* | 5.83 | **8.34** | 5.78 |
| *Chanodichthys erythropterus* | | | 4.39 |
| *Osteochilus salsburyi* | 5.13 | | 5.14 |
| *Cirrhinus molitorella* | 4.10 | | |
| *Sinibotia pulchra* | | **6.37** | |
| Medium dissimilarity | 59.11 | 54.76 | 59.93 |

*3.4. Relationships between Temporal Patterns and Environmental Factors*

The relationships between environmental factors and the ichthyoplankton composition were clarified in the RDA ordination diagram using the data from 28 species and a set of five environmental factors. Accumulated constrained eigenvalues for the first multivariate axes were 0.2467 (RDA1) and 0.0803 (RDA2). After adjustment, $R^2$ changed from 0.41 to 0.39. The *p*-value (ANOVA test) of the full model was sufficiently low, as were the first two canonical axes, which suggested that the samples had been separated well along the axes. The eigenvalues and their contribution to variance are shown in Table 4.

**Table 4.** Summary of the redundancy analysis (RDA). * = $p < 0.05$; *** = $p < 0.001$.

| | RDA1 | RDA2 | RDA3 | RDA4 | RDA5 |
|---|---|---|---|---|---|
| F | 57.68 | 18.67 | 9.3 | 3.81 | 2.31 |
| *p*-value | 0.001 *** | 0.001 *** | 0.01 * | 0.05 * | 0.1 |
| Eigenvalue | 0.07764 | 0.0251 | 0.0125 | 0.0051 | 0.0031 |
| Proportion Explained | 0.62842 | 0.2035 | 0.1013 | 0.0416 | 0.0252 |
| Cumulative Proportion | 0.62842 | 0.8319 | 0.9332 | 0.9748 | 1 |
| Transparency | −0.4736 | 0.0122 | −0.0237 | 0.0112 | 0.0360 |
| DO | −0.1498 | −0.2190 | −0.1180 | −0.0149 | −0.0089 |
| Temperature | 0.3631 | 0.1232 | −0.0737 | 0.0227 | 0.0412 |
| Discharge | 0.4381 | −0.0359 | 0.0862 | 0.0285 | 0.0182 |
| "Water Level" | 0.4483 | 0.0106 | 0.0783 | −0.0122 | 0.0289 |

The RDA biplot of samples and environmental factors (scaling = 1) shows that discharge, temperature, and transparency play important roles in the dispersion of the samples along the RDA1 axis, while DO showed a low contribution to the RDA2 axis (Figure 6a). Moreover, there was a possibility of collinearity between the water level and discharge, but as the variance inflation factor of water level is 10.47, hence it was excluded from further analysis. The plot shows a gradient from left to right, organized into groups, and divided by RDA1 and RDA2. The progression was from the pre-flood period characterized by high transparency and DO, to the flood period with high discharge, and then to the post-flood period with high temperature. The RDA biplot of species and environmental factors shows obvious flood regime assemblage patterns of fish species correlated with different sets of explanatory variables (Figure 6b). At the beginning of summer, the abundance of *S. wui Fang* (Homa) and *S. robusta* (Cobi), is high in the pre-flood period and is associated with higher transparency and higher DO, lower discharge, and temperature. The abundance of *S. argentatus* (Cypr), *S. macrops* (Cypr), *C. erythropterus* (Cypr), *S. dabryi* (Cypr), *C. molitorella*

(Cypr), and *O. salsburyi* (Cypr) is high in the flood period, these species were captured mainly as eggs, and are associated with higher discharge, lower transparency, and DO. *O. mossambicus* (Cich), *O. niloticus* (Cich), and *M. armatus* (Mast) have similar occurrence patterns, all being associated with high precipitation, high discharge, low transparency, and low DO in the flood period, but these species were captured mainly as larval. The abundance of *H. leucisculus* (Cypr), *S. curriculus* (Cypr), and *S. pulchra* (Cobi) are high in the post-flood period, and these are associated with high precipitation. Most other species that are clustered together, show shorter projections on the discharge axis, which indicates that they are more related to the flood.

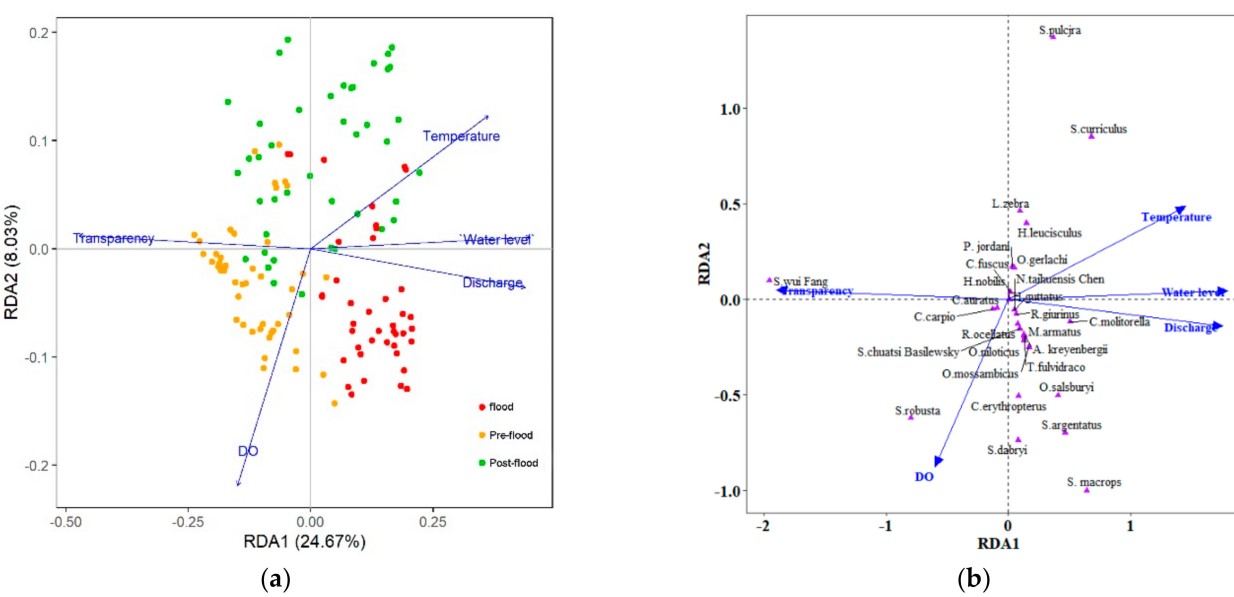

**Figure 6.** Biplot canonical RDA of environmental factors and samples (**a**), and biplot RDA of environmental factors and ichthyoplankton of the main fish species collected in the Laibin section of the Pearl River during the summer of 2017 (**b**).

## 4. Discussion

There is little research regarding the composition and abundance of ichthyoplankton in the upper reaches of the Xijiang River, which is essential information for fishery regulation and management. In this work, ichthyoplankton in the region studied was mainly composed of species from the Cobitidae, Cyprinidae, and Homalopteridae families. This community has changed considerably since 2004–2005 [16]. In a previous investigation into spawning grounds in this area at that time, the dominant species were more often larger-bodied species such as *Mylopharyngodon piceus*, *Ctenopharyngodon idellus*, *Hy. molitrix*, *Hy. nobilis*, *Mystus guttatus*, *Squaliobarbus curriculus*, *Cirrhinus molitorella*, and *Cryrinus carpio*, but all the other dominant species we observed were smaller in size. This shift may be due to anthropogenic alterations in hydrodynamics, since the formation of cascade reservoirs is the main cause of the fish community structure change in the upper reaches of the Xijiang River [29]. This result is similar to that of ichthyoplankton in the upstream of the Three Gorges Dam, the most abundant species were members of Cultrinae, Gobiidae, and Gobioninae [6].

Asian carps were previously the most economically important taxon in this region [16], but we only found *H. nobilis* larval in July. The decline in Asian carp resources is common to the Xijiang River. In the lower reaches, the percentage of these species as larval was 46.6% in 1986 but had declined to 4.6% by 2008 [19]. These species have specific spawning requirements [30], including large, turbid rivers characterized by high turbulence due to hard points or tributary confluences [31]. After the formation of the cascade reservoirs of the upper Xijiang River, the flow regime changed markedly, with a decrease in water velocity and fewer major peak floods [29]. The previous natural lotic environment changed

into a lentic environment. This change may have contributed to the decline in Asian carp spawning.

Generally, the flood regime is the most important force determining seasonality in neotropical rivers. Flood pulses can trigger spawning, especially for fish with drifting eggs [7,12,32]. Longer flood duration and greater flow discharge can result in greater fish production [33]. Water temperature is also thought to play a significant role in fish breeding and population dynamics [34], by stimulating the gonads [35] and affecting spawning frequency [36]. These findings are consistent with our results. This research has shown that ichthyoplankton in the upper reaches of the Xijiang River has significant flood regime variability, with the greatest numbers of most species occurring in the pre-flood or flood period, which can also affect the ecology in the downstream, for example, Chinese white dolphin swim close to estuary during flood seasons due to more abundance of food [37].

It is also obvious that the occurrence of different ichthyoplankton is directly associated with specific environmental requirements. The occurrence of *S. wui Fang* and *S. robusta* climax of the pre-flood period can be explained as follows. In the pre-flood period, higher transparency, higher DO, and low water discharge may be helpful for their spawning, as they lay their eggs in holes under the water to avoid flow disturbances and therefore, need plenty of oxygen. However, the occurrence of *S. argentatus*, *S. macrops*, *C. molitorella*, and *O. salsburyi* climax in the flood period is different. *C. molitorella* and *O. salsburyi* lay their adhesive eggs onto the sand and rocks, the complex hydrogeological conditions may be useful for the adherence of their eggs. As for *S. argentatu* and *S. macrops* egg-laying habits, they lay pelagic eggs and also need complex hydrogeological conditions that will promote the diffusion and drift of their eggs. The selection of reproductive strategies is reflected on the response relationship to environmental factors. Understanding the reproductive strategy of fish, especially dominant species, is the basis of wise decisions on the conservation of fish community diversity.

As mentioned above, the main dominant and common species all lay pelagic eggs and there is a significant correlation between their reproduction and water discharge. Water discharge, in this study, was found to be the main influencing factor. This was underpinned by the fact that the species like to spawn more in the Laibin section, which is the last naturally flowing section of the upper Xijiang. With the operation of Datengxia, the specific flow events and flood pulses will be reduced and the rise of the water level will cause their spawning grounds to be inundated. This may be the most important potential threat to the spawning of species floating fish egg resources in this section. The dominant species showed long reproduction seasons that extended from April to August. The existing fishing ban season lasts until June in the Xijiang every year and does not cover the whole reproduction time. The reproduction of fish needs complete protection against any interference, especially for rare and threatened species. The conservation of rare species by limiting the numbers caught implies that each individual in the population is valuable in the sense that its removal will lead to a further decline in numbers [38]. The same case, the compensatory density dependence is assumed to be very weak in the Yangtze River [6], and the government has ordered a 10-year fishing ban on the Yangtze from 2020, we suggest extending the fishing ban season to cover the entire breeding season. In addition, the large free stretches upstream of the reservoir of the tributary (Liujiang), will probably become important spawning grounds for many fishes. For effective conservation of fish recruitment, the fishery administration should strictly control the fishermen's nets and reduce the occurrence of electric fishing.

In order to provide effective advice and measures for fish conservation, long-term and scientific monitoring is necessary. In this study, we only analyze the occurrence temporal patterns in summer, and the annual occurrence patterns also need to be studied. Meanwhile, the monitoring methods also need to be improved, such as the use of more accurate methods for the identification of fish: DNA barcoding techniques, which has been shown to be accurate to reflect fish larvae and egg diversity [39].

**Author Contributions:** Conceptualization, M.G.; methodology, M.G., Z.W., X.T. and L.H.; investigation, M.G., X.T., J.F. and S.R.; resources, Z.W.; manuscript writing, M.G.; all authors contributed to improving the paper. All authors have read and agreed to the published version of the manuscript.

**Funding:** This work was financially supported by the National Natural Science Foundation of China (32060830).

**Institutional Review Board Statement:** Not applicable.

**Informed Consent Statement:** Studies not involving humans.

**Data Availability Statement:** Some data used during the study are available from the corresponding author by request (Author's email: wuzhiqiang@glut.edu.cn).

**Acknowledgments:** We thank the reviewers for their useful comments and suggestions. We thank the Guangxi Key Laboratory of Environmental Pollution Control Theory and Technology for Science and Education Combined with Science and Technology innovation Base for its assistance with materials and instruments.

**Conflicts of Interest:** The authors declare no conflict of interest.

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
