# Peer review of "Ontogenetic Structure and Temporal Patterns of Summer Ichthyoplankton in Upper Course of the Xijiang River, SW China"

_water, doi:10.3390/w13050703_

Round 1

Reviewer 1 Report

Manuscript number 1115303

Title: Ontogenetic Structure and Temple Patterns of Summer Ichthyoplankton in Upper Reaches of the Xijiang River, SE China.

The authors studied influence of water regime on fish assemblages in the upper Xijiang River ecosystem. The proposed subject is interesting. Studies of habitat disturbance and its effect on fish structure and diversity are very important under human induced decline in global biodiversity.

I have some suggestions for the manuscript before it will be accepted for publication.

Title

Please replace “upper reaches” with “upper course”

“Ichthyoplankton” should be replaced with “fish assemblages” or “fish communities”. Using “plankton” suggests typical open water species.  Results have shown that fish community was represented by different ecological group of fish.

Please explain in what context the word “temple” was used.

Introduction

References to studies on annual and seasonal variations in occurrence patterns of ichthyoplankton, should be added.

The third purpose should be deleted or reformulated, it is not possible to prepare guideline for institution involving in water conservation after one season study.

Results

As it was described in Material and Methods, temperature, water transparency and dissolved oxygen were measured during the study and then used in RDA. Values of these parameters should be presented in sepearte table or figure and described in Environmental factors.

Discussion

There is no references to results of studies focusing on the same life stages (eggs, larval) of fish. Cited studies on duration of reproductive periods, eggs sizes , and parental care migration are not supported by the results of study.

It was pointed that fish community has changed considerably since 2004-2005. Please discuss these differences with dominant species, as with Asian carp.

I suggest some short concluding remarks, to sum up the study.

The manuscript should be verified by native English speaker.

Reviewer 2 Report

The authors did exceptionally well making this study of interest to international readers. It is clear that the text needs extensive editing to correct the English. Below I provide some (but not all) details to improve the ms.

Throughout ms:

'temporal' not 'temple'

use of superscript on units, e.g. m2

Introduction last sentence '2) determine'..... 'control and, 3)'

section 2.1.  Something is wrong with the flow values.. maybe missing 'E', otherwise =2265 m3, which is too low.

section 2.3. was log(x+1) the correct transformation? How was this determined? (I assume log base 10, noting in R log is base e)... also data for PRIMER analyses is usually forth-root transformed. I'm afraid I can't comment further on the tests used as I don't have expertise in such analyses.

What was the other analyses in R?

3.1. Water discharge rate now in m3s-1, which is correct but inconsistent with that in the M&M.

3.2 delete 'raised and'

Table 3. missing caption

Last paragraph page 10 'in The same' 

Reviewer 3 Report

This manuscript examine the temporal variation of fish larvae and eggs in Xijaing river among pre-flood, flood and post-flood period. The pattern is clear and straight forward and result is fine. But the MS is extremely poorly written and must be severely edited before acceptance. Such serious errors includes:

  • What is temple pattern? I assume it is temporal pattern. There are many places appear temple pattern. Please use search function and change all.
  • Legend of table 3 reads “. This is a table. Tables should be placed in the main text near to the first time they are cited.” This is silly error. I think the author should read through and check all such errors in the MS.
  • Many species names have not in italics, especially in discussion. Please check carefully.
  • Pre flood always misspelled as per flood. Please check all.

Major comments as below:

1). Introduction, first paragraph, the author stated “Globally, 77 percent of rivers over 1,000 kilometers in length, are fragmented due to dams, reservoirs, and other facilities [4]. Therefore, it is very necessary to develop practical and effective diversity conservation strategies for fragmented rivers” Fragmented river and also affect genetic differentiation of freshwater organisms. I suggest author to cite the reference below for the effect of fragmented river on affecting the population genetics of freshwater organisms.

Yam, R.S.W and Dudgeon D. (2005). Genetic differentiation of Caridina cantonensis (Decapoda:Atyidae) in Hong Kong streams. J. N. Am. Benthol. Soc., 24(4):845–857

2.) In section 2.1. The authors stated “The upper Xijiang River is located in a geotectonically complex karst area with mountains and valleys on either side, curved channels, beaches, underground streams, and karst caves [11]” This sentence should elaborate these karst caves habitats support specific fauna including cave shrimps. The author should cite the reference below for this statement:

Cai Y, Ng PKL. 2018. Freshwater shrimps from karst caves of southern China, with descriptions of seven new species and the identity of Typhlocaridina linyunensis Li and Luo, 2001 (Crustacea: Decapoda: Caridea). Zool Stud 57:27. doi:10.6620/ZS.2018.57-27

3.) Figure 1 – legend is too simple. The author should elaborate the details including what is the red bar symbol of HEP and white bar symbol HEP under construction.

4). Figure 2 – should indicate which period is pre-flood, flood and post flood in the figure. May be use different shades to highlight the period.

5) Results – should add two additional figures, preferable bar charts. One is the total diversity of species in these three periods. Calculate the Shannon index, number of species of all fish larvae in these three periods. The other bar charts, is based on each family, plot of percentage of eggs and larvae in these three periods. Then discussion should add in these comparisons.

6) SIMPER table 2 is not correct. It is no need to show the percentage value for each periods. Instead, each pairwise comparison – should have a separate table. I assume the set of diagnostic species between each pair of period comparison must be different. Now, this table is certainly not correct.

7) Discussion – third paragraph, the author stated “These findings are consistent with our results. This research has shown that ichthyoplankton in the upper reaches of the Xijiang River has significant flood regime variability, with the greatest numbers of most species occurring in the pre-flood or flood period.” Flood period is very important for the downstream ecology. Flood stream have more fish spawned and more larvae. This also explain why Chinese white dolphin are swim close to estuary river mouth in flood season. The statement above should elaborate “….greatest numbers of most species occurring in the pre-flood or flood period, which can also affect the ecology in the downstream, for example, Chinese white dolphin swim close to estuary during flood seasons due to more abundant of food”. Author can cite:

Li S, Gao H, Hao X, Zhu L, Li T, Zhang H, Zhou Y, Xu X, Yang G, Chen B. 2018. Seasonal, lunar and tidal influences on habitat use of Indo-Pacific humpback dolphins in Beibu Gulf, China. Zool Stud 57:01. doi:10.6620/ZS.2018.57-01.

8) The last paragraph of the discussion, I would suggest to add some suggested improvement on methods of this paper. The present MS use morphological approach to identify the fish larvae. Nowadays, a more common and easy methods is to use DNA barcode. Which has been shown to be accurate to reflect fish larvae and egg diversity. Cite the reference that use DNA barcode for fish larvae below and add in this idea at the last paragraph of discussion.

Chu C, Loh KH, Ng CC, Ooi AL, Konishi Y, Huang SP, Chong VC. 2019. Using DNA barcodes to aid the identification of larval fishes in tropical estuarine waters (Malacca Straits, Malaysia). Zool Stud 58:30. doi:10.6620/ZS.2019.58-30.

Round 2

Reviewer 3 Report

The comments were addressed and the MS can be accepted.

Author Response

We would like to express our sincere thanks to the reviewer for the constructive and positive comments.

Replies to Reviewer

Moderate English changes required

Answer: We have revised the grammar and spelling of the manuscript.
